# Application of Polarization Coulomb Field Scattering to a Physics-Based Compact Model for AlGaN/GaN HFETs with I–V Characteristics

**Yongxiong Yang [1], Yuanjie Lv [2], Zhaojun Lin [1,\*], Guangyuan Jiang [1] and Yang Liu [1]**

[1] School of Microelectronics, Shandong University, Jinan 250100, China; velaa6@163.com (Y.Y.); jgybestabc@163.com (G.J.); ly2451985210@163.com (Y.L.)

[2] National Key Laboratory of Application Specific Integrated Circuit (ASIC), Shijiazhuang 050051, China; yuanjielv@163.com

\* Correspondence: linzj@sdu.edu.cn

**Abstract:** A physics-based model for the output current–voltage (I–V) characteristics of AlGaN/GaN HFETs is developed based on AlGaAs/GaAs HFETs. It is demonstrated that Polarization Coulomb Field (PCF) scattering greatly influences channel electron mobility. With different gate biases, channel electron mobility is varied by PCF scattering. Furthermore, a more negative gate bias and a lower ratio of $l_g/l_{sd}$ (gate length/source-drain space) of the device causes the PCF scattering to have stronger influence on channel electron mobility. This work is the first to apply PCF scattering to a physics-based model for AlGaN/GaN HFETs with I–V characteristics and the results indicate that PCF scattering is essential for a physics-based model to identify I–V characteristics of AlGaN/GaN HFETs.

**Keywords:** AlGaN/GaN HFETs; physics-based model; compact; the PCF scattering; low field carrier mobility

## 1. Introduction

In the last decade, physics-based models of AlGaN/GaN HFETs have been the primary focus of many researchers. Based on calculated surface potential, the ASM model has been developed for industry application [1–4]. At the same time, the traditional physical models derivate from mobility-velocity relation, which has been previously researched and its results studied [5–9]. All physical effects of AlGaN/GaN HFETs, must be included in a physics-based model. However, no existing physics-based model of AlGaN/GaN HFETs has taken Polarization Coulomb Field (PCF) scattering into account, and as a result, some fitting parameters have needed to appear in previous physics-based models of current–voltage (I–V) characteristics, especially in mobility expression [10–12]. Considering that PCF scattering is one of the important physical factors for AlGaN/GaN HFETs, physical-based models of AlGaN/GaN HFETs that fail to take PCF scattering into account can no longer be considered accurate.

PCF scattering is induced by non-uniform distribution of the polarization charges at the AlGaN/GaN interface, and is a unique and integral scattering mechanism in AlGaN/GaN HFETs [13–21]. The low-field mobility of AlGaN/GaN HFETs varies greatly with gate bias, seen in both a more negative gate bias and lower $l_g/l_{sd}$ (gate length/source-drain space) ratio, causing PCF scattering to have a stronger influence on channel electron mobility. PCF scattering is further discussed in Part III. Throughout this article, the continuous heterostructure field-effect-transistor model referenced in the study by Turner et al. [7] is adopted due to its direct relationship to low field mobility, and PCF scattering is applied in the physical-based model's I–V characteristics and investigated by modeling AlGaN/GaN HFETs' I–V characteristics.

## 2. Model

### 2.1. Linear Region

Our study is based on a charge-control model [7]. The velocity-field characteristics are as follows:

$$
v = \begin{cases} \dfrac{\mu F_x}{1+\frac{F_x}{2F_s}}, & for\ F_x \le 2F_s \\[3mm] v_{sat}, & for\ F_x \ge 2F_s \end{cases} \tag{1}
$$

where $v$ is the electron drift velocity, $\mu$ is the low-field mobility, $F_x$ is the longitudinal electric field along the channel, $F_s\ (= v_{sat}/\mu)$ is the characteristic field of the velocity saturation, and $v_{sat}$ is the electron saturation velocity. The drain-to-source current in the linear region $I_{DSL}$ is given by reference [7]:

$$
I_{DSL} = \frac{1}{R_n}\frac{2V_{GT}V_{DS}-V_{DS}^2}{V_{DS}+2V_L} \tag{2}
$$

In (2), $R_n$ ($R_n = 1/(WC_i\, v_{sat})$) is the intrinsic device transresistance in the saturation region at the short channel limit. $C_i$ is gate-to-channel capacitance. ($V_{GT} = V_{GS} - V_T$) is the effective gate bias for the intrinsic device. $V_{GS}$ is the intrinsic gate-to-source voltage. $V_{DS}$ is the intrinsic drain to source voltage. $V_L = F_s \times L$, $L$ is the effective electrical gate length. $F_s$ is the characteristic field of the velocity saturation. By adapting parasitic resistance ($V_{GS} = V_{gs} - IR_S$, $V_{DS} = V_{ds} - I(R_S + R_D)$) and integrating from source to drain, the approximate expression for the current–voltage characteristics of the GaN HFETs in a linear region is represented below [7]:

$$
I_{DSL} = \frac{2V_{gt}V_{ds}-V_{ds}^2}{A+\sqrt{A^2-B}} \tag{3}
$$

in which:

$$
A = \left(\frac{R_n}{2}-R_D\right)V_{ds}+(R_S+R_D)V_{gt}+R_nV_L \tag{4}
$$

and

$$
B = (R_s+R_D)(R_S-R_D+R_n)\left(2V_{gt}V_{ds}-V_{ds}^2\right) \tag{5}
$$

Here, $R_s$ denotes the parasitic source resistance, $R_D$ the parasitic drain resistance, $V_{gt}$ the effective gate bias for the extrinsic device, and $V_{ds}$ the extrinsic drain-to-source voltage.

### 2.2. Saturation Region

When $V_{ds}$ surpasses saturation voltage $V_{dsat}$, the channel can be segregated into two regions. Region 1 represents when the channel potential is lower than the saturation potential, and Gradual Channel Approximation is valid. Region 2 represents when the electron velocity is equal to the saturation velocity; under these circumstances, Gradual Channel Approximation is no longer relevant due to rapid variation of both transverse and longitudinal components of the electric field [7], and the saturation current is provided by Tunner [7],

$$
I_{DSS} = I_{DSAT}\left\{1+\frac{2V_\lambda}{V_{gt}+2V_L-I_{DSAT}R_s}In[1+K]\right\} \tag{6}
$$

$$
K = \frac{(V_{ds}-V_{dsat})(V_{gt}-2V_L-I_{DSAT}R_s)^2}{8V_LV_\lambda\left(V_{gt}+V_L-I_{DSAT}R_s\right)} \tag{7}
$$

where $V_\lambda = F_s \times \sqrt{t \times d}$, $d$ is AlGaN layer thickness, and $t$ is effective thickness of the two-dimensional electron gas in the saturated region.

## 3. PCF Scattering

Non-uniform polarization charges at the AlGaN/GaN interface are the source of PCF scattering. Spontaneous and piezoelectric polarization cause the polarization charges at the AlGaN/GaN interface to be uniform prior to device processing, as shown in Figure 1. Uniform distribution of polarization charges is unable to scatter the channel electron. When the device is working on a different gate bias, polarization charges under the gate vary with gate bias due to the converse piezoelectric effect. Considering that polarization charges in gate-to-source and gate-to-drain regions do not change with gate bias, polarization charges distribute non-uniformly when the gate bias is changed.

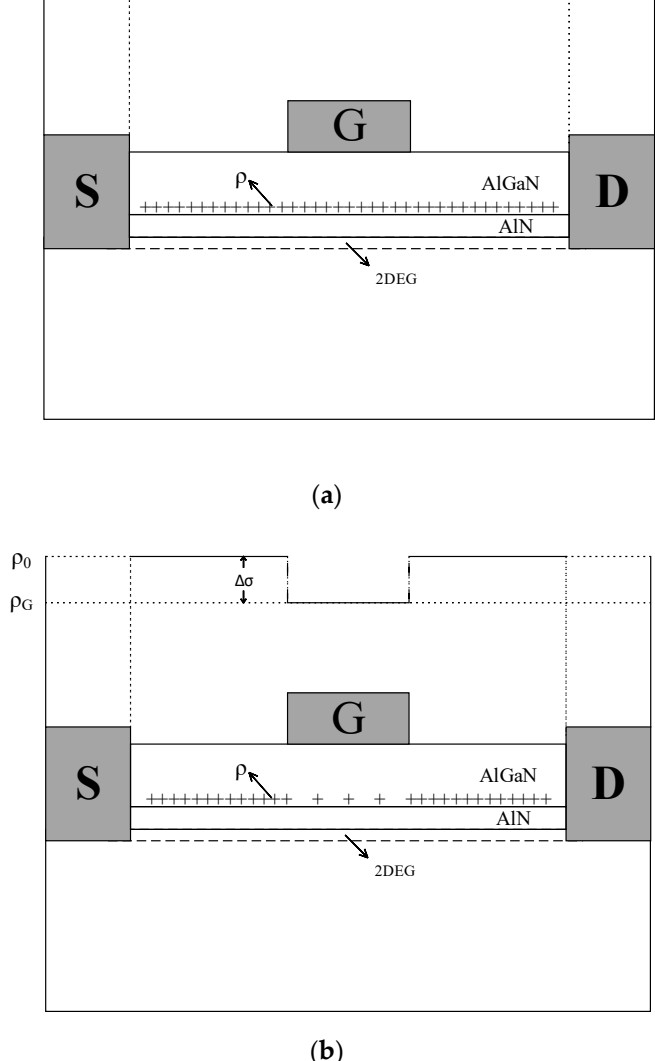

(**a**)

(**b**)

**Figure 1.** The polarization charge distribution of AlGaN/GaN HFETs (**a**) when the gate bias is 0; (**b**) when the gate bias is negative. ($\rho_0$ is the polarization charges in equilibrium state which equals the polarization charges in 0 gate bias, $\rho_G$ is the polarization charges under the gate.)

Perturbation theory has also been adopted in PCF scattering calculations. Difference between the polarization charges under the gate and under the gate-source/gate-drain region $\Delta\sigma$ creates a perturbation potential to the channel electrons under the gate. When gate bias is applied to the AlGaN barrier layer, the difference between polarization charges under the gate and under the gate-source/gate-drain region $\Delta\sigma$ can be modulated, and the perturbation effect changes

accordingly. Then, the Fermi Golden Rule is applied to calculate the energy-dependent scattering rate for PCF scattering:

$$\frac{1}{\tau_{PCF}(E)} = \frac{Am^*}{2\pi\hbar^3} \int_{-\pi}^{\pi} |\frac{M_{k\to k'}}{S(q, T_e)}|^2 (1 - cos\theta) d\theta \tag{8}$$

where $\theta$ is the scattering angle between the final state $\boldsymbol{k'}$ and initial state $\boldsymbol{k}$, $m^*$ is the effective electron mass and $\hbar$ is the Planck constant. The screening function $S(q, T_e)$ is

$$S(q, T_e) = 1 + \frac{e^2 F(q)\Pi(q, T_e, E)}{2\varepsilon_0 \varepsilon_s q} \tag{9}$$

where $e$ is the electron charge, $\varepsilon_0$ is the vacuum dielectric constant, $\varepsilon_s$ is the relative dielectric constant, the form factor $F(q)$ is

$$F(q) = \int_0^\infty \int_0^\infty \psi^2(z)\psi^2(z\prime)exp(-q|z - z'|)dzdz\prime \tag{10}$$

and the polarizability function $\Pi(q, T_e, E)$ is

$$\Pi(q, T_e, E) = \frac{m^*}{4\pi\hbar^2 k_B T_e} \times \int_0^\infty \frac{1 - \Theta(q - 2k_F)[1 - (2k_F/q)^2]^{1/2}}{cosh^2\left[\frac{E_F - E}{2k_B T_e}\right]} dE \tag{11}$$

In the equation above, $\Theta(x)$ is the usual step function, $k_F = (2\pi n_{2-D})^{1/2}$ is the Fermi wave vector, $T_e$ is the electron temperature, $E_F$ is the Fermi energy, and $E$ is the energy.

The 2-D electron wave function can be expressed as $\Psi(x, y, z) = A^{-\frac{1}{2}}\psi(z)\exp(ik_x x + ik_y y)$, where $A$ is the 2-D normalization constant. Therefore, the transition matrix can be written as:

$$M_{k\to k'} = A^{-1} \int_0^\infty \psi_{k'}^*(z)\left[\int_{-\frac{L_G}{2}-L_{GS}}^{-\frac{L_G}{2}} dx \int_0^W V(x, y, z) \times \exp(-iq_x x - iq_y y)dy\right]\psi_k(z)dz = \\ A^{-1} \int_0^\infty \psi_{k'}^*(z)\left[V(q_x, q_y, z)\right]\psi_k(z)dz \tag{12}$$

The $q_x$ and $q_y$ are the components of $q$ in the x-direction and y-direction, respectively. $\boldsymbol{q} = \boldsymbol{k'} - \boldsymbol{k}$ refers to the change of wave vector in the scattering process.

The PCF scattering potential $V(x, y, z)$ can be written as:

$$V(x, y, z) = -\frac{e}{4\pi\varepsilon_s\varepsilon_0}\int_{-L_{GS}-\frac{L_G}{2}+l}^{-\frac{L_G}{2}} dx' \times \int_0^W \frac{\Delta\sigma\left(x' + \frac{L_G}{2} + L_{GS} - l\right)}{(L_{GS}-l)\sqrt{(x-x')^2 + (y-y')^2 + z^2}} dy' - \\ \frac{e}{4\pi\varepsilon_s\varepsilon_0}\int_{\frac{L_G}{2}}^{L_{GD}+\frac{L_G}{2}+l} dx' \times \int_0^W \frac{\Delta\sigma\left(x' - \frac{L_G}{2} - L_{GD} + l\right)}{(L_{GD}-l)\sqrt{(x-x')^2 + (y-y')^2 + z^2}} dy' \tag{13}$$

As we seen from Equation (13), the absolute value of PCF scattering potential $V(x, y, z)$ increases with the difference value of the polarization charges under the gate and under the gate-source/gate-drain region $\Delta\sigma$, and $\Delta\sigma$ increases with the gate bias negatively increasing. Therefore PCF scattering is greater when the gate bias is more negative.

As seen in Figure 1, when the gate bias is negative, the additional polarization charge $\Delta\sigma$ in the gate-source and gate-drain regions scatters the electron under the gate. Anytime the length of the gate to source and gate to drain is larger, or the gate bias is more negative, the additional polarization charge $\Delta\sigma$ will increase, resulting in the PCF scattering intensifying. The reduction in gate length will narrow the distance between the additional polarization charge $\Delta\sigma$ and the electron under the gate, also causing PCF scattering to intensify. In summary, the lower ratio of $l_g/l_{sd}$ and a more negative gate bias both lead to stronger PCF scattering.

As seen in Equations (4) and (6), $\mu$ is a vital parameter for both linear and saturation regions contained in $V_L = F_S \times L = \frac{v_{sat}}{\mu} \times L$. All other parameters remain the same in any device. Therefore, by investigating the proposed model of AlGaN/GaN HFETs with different low-field electron mobility, we can identify the impact of PCF scattering on the AlGaN/GaN HFETs model.

## 4. Experiment

The samples were grown via molecular beam epitaxy (MBE) on a SiC GaN cap layer, a 25.5 nm $Al_{0.21}Ga_{0.79}N$ barrier layer, a 0.7 nm AlN interlayer, a 1 µm undoped GaN layer, a 1 µm C-doped GaN buffer layer, and a 100 nm AlN nucleation layer. The epitaxial structure is shown in Figure 2. The device fabrication started with mesa isolation, formed by inductively coupled plasma reactive ion etching (ICP-RIE). Then, the source and drain ohmic contacts were formed by depositing Ti/Al/Ni/Au multilayer and annealing at 850 °C for 30 s in nitrogen atmosphere. The Ni/Au gate was fabricated and located in the middle of the drain and source electrodes. Finally, the devices were passivated by using a 50 nm SiN deposited by PECVD. Hall measurement led to a 2DEG density of $8.4 \times 10^{12}$ cm$^{-2}$ and an electron mobility of 2030 cm$^2$/V·s. The specific resistivity of 0.85 Ω·mm was derived by TLM (Transmission Line Method). For device size, the gate width ($W$) is 80 µm, and the gate-source spacing ($L_{GS}$) and the gate-drain spacing ($L_{GD}$) are both 1 µm. The devices have gate lengths ($L_G$) of 0.25 µm and 0.5 µm, which were marked as sample 1 and sample 2, respectively. The area of the ohmic region in the source side is four times the area of the ohmic region in the drain side, hence $R_s$ is smaller than $R_D$.

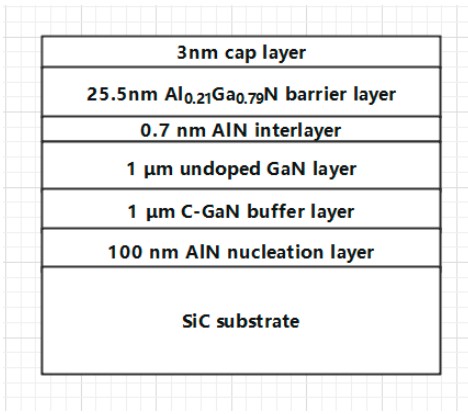

**Figure 2.** Schematic structure of AlGaN/GaN HFET.

## 5. Comparison of Experimental and Calculated Result

When PCF scattering is applied to output I–V curve calculation, the low-field electron mobility µ corresponding to different gate biases can be calculated and obtained. Resulting PCF scattering rates are calculated by Equation (8), and any calculations of other scattering like POP, PE and so on can be referenced in Cui et al. [15–18]. The calculated parameters and the related parameters are both listed in Tables 1–3. For sample 1 and sample 2, low-field electron mobility µ was decreased from 1720 cm$^2$/V·S and 1748 cm$^2$/V·S (corresponding to $V_{gs}$ = 0 V) to 889 cm$^2$/V·S and 1065 cm$^2$/V·S (corresponding to $V_{gs}$ = −3 V), respectively. The larger negative gate bias leads to smaller electron mobility due to tremendous PCF scattering, caused by the larger additional polarization charges caused by the gate bias. Note that sample 1 and sample 2's low-field $R_s$ and $R_D$ corresponding to different gate biases are less varied, the reason being that the gate length to the source-drain spacing ratio for both samples are small, and the influence of PCF scattering on the $R_s$ and $R_D$ is weak [16–18].

**Table 1.** Parameters in simulation.

| Symbol | Meaning | Quantity | 0.25 μm Gate Length Device | 0.5 μm Gate Length Device |
|---|---|---|---|---|
| $C_i$ | Gate-to-channel capacitance | F/m$^2$ | $3.45 \times 10^{-3}$ | $3.45 \times 10^{-3}$ |
| $v_{sat}$ | Electron saturation velocity | cm/s | $4.02 \times 10^6$ | $6.5 \times 10^6$ |
| $F_s$ | Characteristic field of the velocity saturation | V/m | $2.05 \times 10^5$ | $3.72 \times 10^5$ |
| $V_t$ | Threshold voltage | V | $-3.9$ | $-3.9$ |
| $n_{2\text{-}D}$ | Two-dimensional electron density in zero bias | m$^{-2}$ | $8.4 \times 10^{16}$ | $8.4 \times 10^{16}$ |

**Table 2.** Mobility, gate-to-source resistance and gate-to-drain resistance of sample 1.

| Gate Bias/V | Mobility/cm$^2$/V·s | $R_S$(Including Ohmic Resistance)/Ω | $R_D$(Including Ohmic Resistance)/Ω |
|---|---|---|---|
| 0 | 1720 | 10.28 | 24.41 |
| −0.5 | 1683 | 10.22 | 24.34 |
| −1 | 1611 | 10.25 | 24.38 |
| −1.5 | 1553 | 10.18 | 24.30 |
| −2 | 1418 | 10.17 | 24.29 |
| −2.5 | 1220 | 10.18 | 24.30 |
| −3 | 889 | 10.69 | 24.86 |

**Table 3.** Mobility, gate-to-source resistance and gate-to-drain resistance of sample 2.

| Gate Bias/V | Mobility/cm$^2$/V·s | $R_S$(Including Ohmic Resistance)/Ω | $R_D$(Including Ohmic Resistance)/Ω |
|---|---|---|---|
| 0 | 1748 | 10.02 | 24.13 |
| −0.5 | 1721 | 10.13 | 24.21 |
| −1 | 1667 | 10.08 | 24.18 |
| −1.5 | 1607 | 10.12 | 24.21 |
| −2 | 1476 | 10.20 | 24.27 |
| −2.5 | 1287 | 10.33 | 24.38 |
| −3 | 1065 | 11.60 | 25.46 |

The saturation point $V_{dsat}$ at zero gate bias is extracted by using a combined graphing method by Turner et al. [7], Equations (3) and (14).

$$V_{dsat} = \frac{2\left(V_{gt} - I_{DSAT}R_S\right)V_L}{\left(V_{gt} - I_{DSAT}R_S + 2V_L\right)} \tag{14}$$

Here it is shown how electron saturation velocity $v_{sat}$ and saturation current $I_{DSAT}$ are obtained for zero gate bias. Electron saturation velocity is the same in different gate biases, so the saturation point $(V_{dsat}, I_{DSAT})$ at the other gate biases is determined by simultaneously solving Equations (3) and (14), in which electron saturation velocity $v_{sat}$ is checked along with the value for zero gate bias. Specifically, when electron velocity at the channel end point reaches saturation velocity, the corresponding $V_{ds}$ is $V_{dsat}$. Threshold voltage is determined by the transfer characteristic curve. With the calculated I–V curve, Equation (3) is applied in the linear part, and Equation (6) is applied in the saturation part.

Figures 3 and 4 compare the measured and calculated curves. The accuracy between the calculated and measured curves from the figures is outstanding for 0.25 μm and 0.5 μm gate length devices in regard to PCF scattering, and accuracy with PCF scattering is much better than without PCF scattering both in the linear and saturation regions, especially in sample 2 with the lower $l_g/l_{sd}$ ratio with a gate length of 0.25 μm.

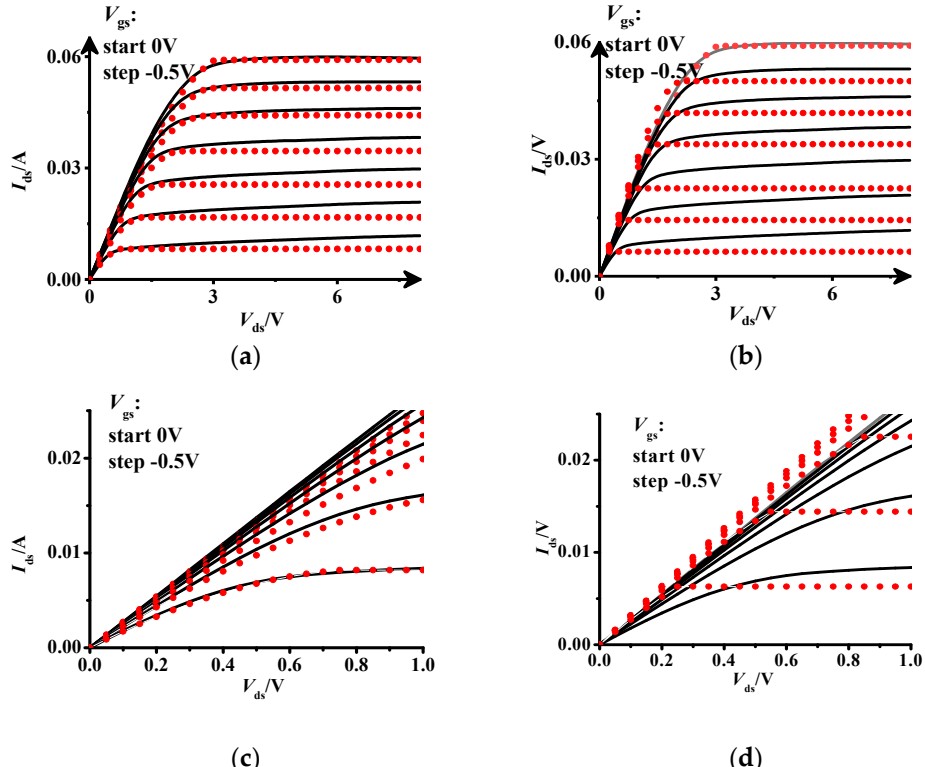

**Figure 3.** The 0.25 um gate length device $V_{ds}$ and $I_{ds}$ relation under different gate biases (**a**) concerning Polarization Coulomb Field (PCF), (**b**) without concerning PCF, (**c**) linear region concerning PCF, (**d**) linear region of without concerning PCF, Solid line: calculated curve, dot line: measured curve.

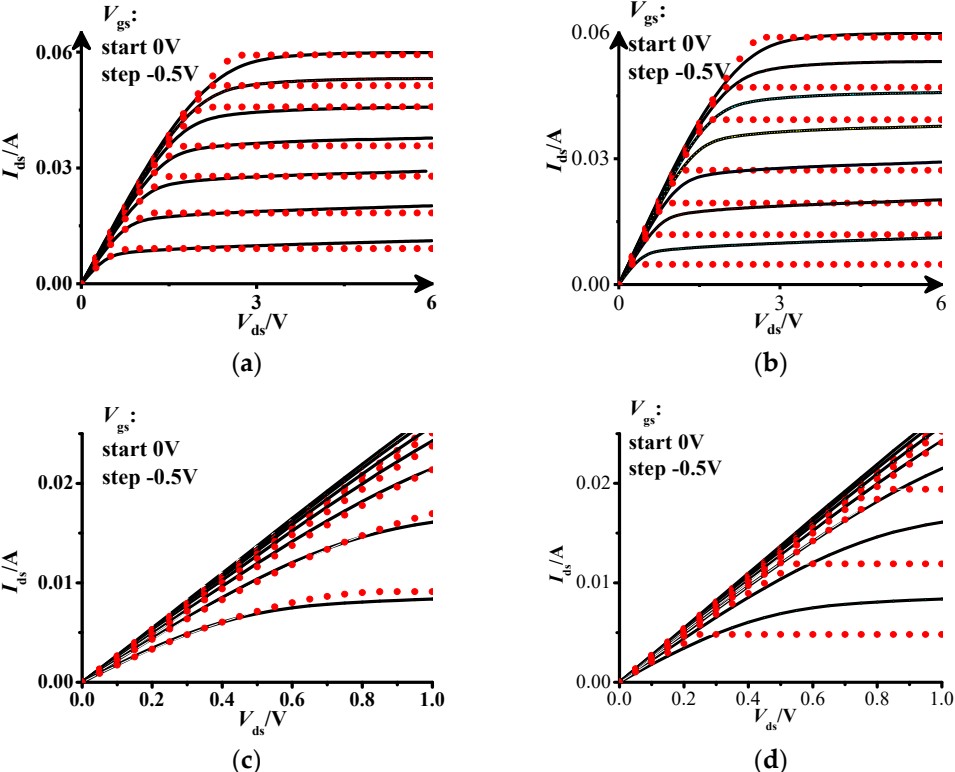

**Figure 4.** The 0.5 um gate length device $V_{ds}$ and $I_{ds}$ relation under different gate biases (**a**) concerning PCF, (**b**) without concerning PCF, (**c**) linear region concerning PCF, (**d**) linear region without concerning PCF. Solid line: measured curve, dot line: calculated curve.

When PCF scattering is irrelevant, the low-field electron mobility $\mu$ for sample 1 and sample 2 is constant among different gate biases. As shown in Figures 3 and 4, when the gate bias is zero, the calculated results both with and without PCF scattering compare favorably with the testing results, due to no additional polarization charges when the gate bias is 0, and PCF scattering can therefore be ignored. With different negative gate biases, the channel current concerning PCF scattering in the linear region matches well with the testing results found for sample 1 and sample 2. However, when PCF scattering could be ignored, the calculated channel current is significantly higher than the testing current, and this phenomenon is more apparent when gate bias is even more negative as shown in (c,d) of Figures 3 and 4. The cause of this phenomenon is that in the linear region, when PCF scattering is irrelevant, the channel electron mobility under the gate is higher than the real value, leading to a higher channel current. The calculated channel currents in saturation regions in sample 1 and sample 2 are much lower than the testing results when PCF scattering can be ignored, due to the ease in which the electron saturation velocity and saturation point are reached.

## 6. Conclusions

The influence of PCF scattering in physics-based models for the output I–V characteristics of submicro AlGaN/GaN HFETs has been investigated based on the charge-control model and its results have been analyzed throughout this paper. The established model matches the experimental results accurately by taking PCF scattering into account. While the result is unacceptable when PCF scattering is ignored, the phenomenon is more significant when the $l_g/l_{sd}$ ratio is lower. In summary, it can be reasonably determined that PCF scattering is essential for any accurate physics-based model's I–V characteristics of AlGaN/GaN HFETs.

**Author Contributions:** Y.Y. put forward ideas and designed the experiment; G.J. and Y.L. (Yuanjie Lv) analyzed the data; Y.L. (Yang Liu) performed the experiments; Z.L. proofread the paper and supervised the overall work. All authors have read and agreed to the published version of the manuscript.

**Funding:** This work was funded by the National Natural Science Foundation of China under Grant Nos. 11974210 and 11574182.

**Conflicts of Interest:** The authors declare no conflict of interest.

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
