# Peer review of "Application of Polarization Coulomb Field Scattering to a Physics-Based Compact Model for AlGaN/GaN HFETs with I–V Characteristics"

_electronics, doi:10.3390/electronics9101719_

Round 1
Reviewer 1 Report
According to the Reviewer’s opinion, this manuscript presents some interesting features and might be worthy of publication in Electronics. Nevertheless, revisions and clarifications are required to make this manuscript publishable.
In particular, the Authors have to strengthen the experimental validation.
As the GaN HEMT technology is aimed at high-power applications, it would be interesting to apply the proposed approach also to a GaN HEMT with a gate width much larger than only 80 um and at a drain-source voltage much higher than only 8 V.
This study would gain much more values by adding frequency dependent measurements (e.g., scattering parameters).
For the sake of completeness, the Authors should mention in the Introduction about the well-known Angelov equations that have been widely adopted in the literature for modeling IV characteristics for the GaN technology (e.g., G. Avolio et al. “Waveforms-only based nonlinear de-embedding in active devices” MWCL 2012, G. Avolio et al. “Identification technique of FET model based on vector nonlinear measurements”, Electron. Lett 2011).
The Authors should comment on the low-frequency effects playing a crucial and critical role for the microwave GaN HEMT technology and a few relevant references should be given.
Please avoid capitalizing the first letters of seemingly random words in sentences.
The Authors should stick to the reference style of the journal (e.g., it seems that sometimes only the first author is reported in the authors’ list).
Reviewer 2 Report
- Authors have already published a paper in AIP Advances (https://doi.org/10.1063/5.0012615). Reviewer feels that this work is almost same as the one published in the AIP advances. Reviewer noticed that this work is discussed the HEMT structure on Sapphire substrate. Authors did not cited the AIP paper.
- If this work is different from AIP advances paper, Authors has to provide valid explanation.
- Typically, AlGaN/GaN HEMT structure comes with GaN cap layer. This is to avoid the oxidation of AlGaN. Will the GaN cap layer influences the incorporation of PCF? This has to be addressed in the revised manuscript
- The device passivation is also affects the polarization. Authors did not discussed this point. Why?
- Authors has given Specific resistivity of 0.85 ohm-mm. Normally, this is called as contact resistance. Authors has to correct this in the revised manuscript.
- Authors are also required to provide the thicknesses of the grown api-layers along with the 2DEG properties.
- This paper is also not discuss about the self-heating. Why this behaviour is taken as an input for device modelling. Why?
- Authors has to run the spell check (Example: "Submicro" in the conclusion) and the improvement in the English Language is also required.
Reviewer 3 Report
GaN-based devices have been simulated extensively in the past. The state of the art consists in Monte Carlo simulations with scattering rates calculated using ab initio methods and extensive comparison with experiments.
Unfortunately, in this manuscript the authors employ sub-par simulation methods with an equally sub-par physical model and do not even cite the relevant literature, most notably the work performed by professor Bellotti's group at Boston University (for example, J. Fang et al., Phys. Rev. Appl. 11, 044045 (2019);A. Kyrtsos, M. Matsubara, and E. Bellotti, Phys. Rev. B 99, 035201 (2019), just to mention a few very recent publications).
Therefore, this manuscript does not add anything to our knowledge of electron transport in GaN or AlGaN/GaN-based FETs. Even the effect of scattering induced by the polarization field -- effect that is emphasized in this manuscript -- has already been studied extensively in the past (see, for example, Ming Yang et al., IEEE TED 63, 1471 (2016); P. Ciu et al., IEEE TED 64, 1038 (2017) and Sci. Rep. 8 12850 (2018); G. Jiang et al., AIP Advances 10, 075212 (2020))).
In conclusion, regretfully, I cannot recommend this manuscript for publication.
Round 2
Reviewer 1 Report
The revised manuscript might be now suitable for publication in Electronics.
Author Response
Thank you very much for your valuable comment. We treasure these comments very much. They do improve our paper and our knowledge on this topic a lot.
Reviewer 2 Report
- Though the authors covered most of the reviewer's comments, the comment #3's answer is having discrepancy with the revised manuscript. Authors has to include the Schematic diagram along with the revised process conditions in the revised manuscript.
- The stated contact resistance values are different from the submitted manuscript. Why? Authors are required to revise manuscript with the correct, 2DEG properties and its process conditions (including contact resistance values.
Comment 3:
Authors are also required to provide the thicknesses of the grown api-layers along with the 2DEG properties.
Our explanation:
Thank you very much for your valuable comment. We think it is a very valuable suggestion for improving our manuscript.
The epi-layer information is as follows. This chip is on the same wafer with Cui’s [6], and is fabricatedby Cui, and Lv. Detailed information can be seen in Cui’s paper. The samples were grown via molecular beam epitaxy (MBE) on a SiC substrate. The epitaxial structure, as shown in Fig. 1, contains a 3 nm The samples were grown via molecular beam epitaxy (MBE) on a SiC GaN cap layer, a 25.5 nm Al0.21Ga0.79N barrier layer, a 0.7 nm AlN interlayer, a 1 μm undoped GaN layer, a 1 μm C-doped GaN buffer layer, and a 100 nm AlN nucleation layer. The sheet electron concentration and electron mobility obtained from Hall measurements were 8.4×1012cm2
and 2340 cm2/V•s, respectively. The device fabrication started with mesa isolation, formed by inductively coupled plasma reactive ion etching (ICP-RIE). Then the source and drain ohmic contacts were formed by depositing Ti/Al/Ni/Au multilayer and annealing at 850℃ for 30s in nitrogen atmosphere. The transmission line measurement (TLM) showed that the ohmiccontact resistance was 1.07 Ω⋅mm. Ni/Au gate was fabricated and located in the middle of the drain and source electrodes. Finally, the devices were passivated by using a 50nm SiN deposited by PECVD. As shown in Fig. 1, the gate-drain distance LGDwas 1 μm, the gate-source distance LGS was 1 μm. The current-voltage (I–V) measurements were carried out at room temperature by using an Agilent B1500A semiconductor parameter analyzer.
Reviewer 3 Report
I am not sure what to think of the authors' reply and their dismissive tone. The authors have completely ignored/disregarded my previous suggestions and remarks stating, instead, that since the industry needs "urgently" a compact model for GaN-based devices, one should not worry about any suggestions for improving the physical models or even for comparing their model to what better models can do. I am a loss for words. Clearly, the authors present a "compact" model (a crucial word that is completely absent in the manuscript and, instead, must appear in the title itself). In this case, I must leave it to the editors to decide what to do with a manuscript that is essentially devoid of any physics, that does not improve our understanding of GaN-based devices and focuses, instead, on an industrial application whose usefulness is highly doubtful. Is such a work suitable for publication in this journal? Or would be more appropriate as a contribution to some VLSI conference?
